# Motivating Diabetic and Hypertensive Patients to Engage in Regular Physical Activity: A Multi-Component Intervention Derived from the Concept of Photovoice

**DOI:** 10.3390/ijerph16071219

**Published:** 2019-04-05

**Authors:** Angela Y.M. Leung, Pui Hing Chau, Isaac S.H. Leung, Michael Tse, Perick L.C. Wong, Wai Ming Tam, Doris Y. P. Leung

**Affiliations:** 1Centre for Geronotological Nursing (CGN), School of Nursing, The Hong Kong Polytechnic University, Hong Kong, China; isaac.sh.leung@polyu.edu.hk; 2School of Nursing, The University of Hong Kong, Hong Kong, China; phchau@graduate.hku.hk; 3Institute of Human Performance, The University of Hong Kong, Hong Kong, China; matse@hku.hk; 4Hong Kong Sheng Kung Hui Welfare Council Western District Elderly Community Centre, Hong Kong, China; lcwong@skhwc.org.hk (P.L.C.W.); wmtam@skhwc.org.hk (W.M.T.); 5School of Nursing, The Hong Kong Polytechnic University, Hong Kong, China; doris.yp.leung@polyu.edu.hk

**Keywords:** photovoice, chronic illness, physical activity, barriers, facilitator

## Abstract

*Aims:* A community-based multi-component intervention (increasing awareness of the importance of physical activity in chronic illness management through reading comic books, training regarding warm-up stretching exercises, identifying facilitators and barriers to exercise through photosharing, supporting self-reflection and development of action plans) was developed to promote physical activity (PA) among patients with diabetes and hypertension. This study aimed to evaluate the efficacy of this intervention on health behaviour (walking) and health outcomes. *Design:* A non-randomized controlled trial with waitlisted control and pre- and post-measures. *Setting:* Community centres for the elderly. *Participants:* A total of 204 older adults with diabetes and/or hypertension were recruited. They were assigned to either the intervention group (IG) or waitlisted to the control group (CG). *Intervention:* Under the supervision of a nurse, six weekly group meetings were arranged in community centres for the elderly in which the participants freely exchanged their views regarding the barriers and facilitators of regular physical activity. Participants were encouraged to take photos in their neighbourhood or at home and brought these photos to share at the group meetings. The photos showed both the barriers and the facilitators to PA. In the last meeting, each participant worked out a plan to perform PA in the coming four weeks. *Measures:* PA referred to the number of steps taken per day and it was measured by a Garmin Accelerometer at baseline, Week 6 and Week 10. Other measures included the nine-item Self-Efficacy Scale for Exercise–Chinese version (SEE-C), and the 23-item Chinese Barriers to Exercise Scale and Senior Fitness Tests. Generalised estimating equations (GEE) models compared the outcomes over time between IG and CG. *Results:* A statistically significant difference in the changes in the average number of steps taken daily between the two groups at Week 10 (mean difference = 965.4; 95% confidence interval: 92.2, 1838.6, *p* = 0.030) was observed, although the difference at Week 6 was non-significant (mean difference = 777.6; 95% confidence interval: −35.3, 1590.5, *p* = 0.061). IG participants also showed significant improvements in lower body strength (mean difference = 0.967; 95% confidence interval: 0.029, 1.904, *p* = 0.043) and lower limb flexibility (mean difference = 2.068; 95% confidence interval: 0.404, 3.731, *p* = 0.015) at Week 10 compared to CG participants. *Conclusion:* This multi-component intervention improved the participants’ physical activity level and physical fitness, particularly in lower limb flexibility and body strength.

## 1. Introduction

Photovoice, a health literacy tool, has received growing attention in health promotion and health education since its development in the mid-1990s [1]. A method such as this encourages participants to explore and consider how the environment affects one’s health behaviour and share their views. Using photos or pictures, participants reflected on their own lifestyles, identified the pros and cons of their existing behaviour, and eventually fostered a positive attitude towards healthy lifestyles and made necessary changes to their behaviour [2,3,4]. Photovoice has been used in various studies in a range of populations to resolve different health problems, for example, promoting the practice of safe sex and preventing the spread of the sexually transmitted disease, human immunodeficiency viruses (HIV) [3], promoting physical activities among rural youth and urban older adults [4,5], advocating obesity prevention, healthy food supply and a safe environment for walking in the community [6], acquiring and preparing food in consideration of financial and environmental limitations [7], connecting new immigrants’ health to social determinants [8], and more recently, empowering vulnerable older adults with heart failures [9], understanding the dietary behaviours of older Filipino people with cardiovascular disease [10], and enhancing patient involvement in palliative care planning [11]. After reviewing 37 papers on the use of photovoice in different health promotion settings, it was also concluded that photovoice contributed to a better understanding of individuals’ behaviour and explained the relationship between behaviour and sociocultural context [12]. With the existing evidence, we acknowledge photovoice as a powerful tool to empower individuals by giving them opportunities to talk about their individual problems (for example, having certain barriers to physical activities) and find ways to solve these problems. Through this process, empowerment occurred and this will support individual growth (for example, participants may consider making a change in their behaviour). In the previous studies, photovoice was considered as a qualitative method to collect participants’ views on a health-related topic and, subsequently, their comments helped to generate policy recommendations. However, no study has ever investigated the actual effect of this kind of photo-oriented group discussion on health behaviours and the possible consequences on the individuals per se (changes in health outcomes after behavioural changes). Therefore, we deliberately formulated a six-week community-based health promotion programme (or the intervention) with photovoice as one of the components in the intervention. This intervention did not solely use photos and group discussion, it also comprised other components such as understanding the importance of physical activity in chronic illness management through the reading of comic books and developing action plans for the next four weeks in the neighbourhood. The innovative aspect of this study is the incorporation of the concept of photovoice in an intervention as we believe that the participants would achieve personal growth and make decisions regarding their behaviour after their participation in the intervention.

For the elderly population with type II diabetes and/or hypertension, maintaining physical activity at a recommended level (for example, 30 min of moderate exercise five times per week or for 150 min per week) [13] is of high priority in disease management. We target this population because physical activity at the recommended level plays an important role in glycemic control in patients with type II diabetes [13] and helps maintain normal blood pressure for hypertensive patients [14]. However, in reality, there are many older adults with hypertension and/or type II diabetes who are often overweight and have other comorbidities who fail to achieve the recommended physical activity level. The reasons for this include a lack of guidance and the physical weakness of type II diabetic patients [15].

Using the concept of photovoice [12], the project team developed a health promotion programme titled ‘Make a Change through Photovoice (MCPv): engaging diabetic and hypertensive patients in physical activity’. This programme aimed to motivate the older adults with diabetic and/or hypertension to undertake physical activity regularly by identifying facilitators and barriers to physical activity. ‘Facilitators’ refer to the factors that made the participants engage in physical activity while ‘barriers’ refer to the factors that deterred the participants from doing favourable types of physical activity [7]. We hypothesised that this intervention would increase the participants’ physical activity level, physical fitness and self-efficacy to do exercise.

## 2. Materials and Methods

### 2.1. Design and Participants

This was a non-randomized controlled trial with a waitlist control and a pre- and post-design. Participants were recruited from four community centres for the elderly in the Western District of Hong Kong Special Administrative Region, China, and the recruitment notices were disseminated via the centre newsletters, monthly meetings and promotional booths in the neighbourhood. Recruitment of participants was divided into 25 batches from July 2014 to November 2015. For each batch of recruitment, we recruited only 9–12 participants. Such an arrangement was made because we had a limited number of digital cameras that could be loaned to the participants. Posters/leaflets were delivered in community centres to encourage their members to join a health seminar. The seminar introduced the programme and recruited participants after a short talk related to diabetes and exercise. The inclusion criteria of the participants were: (1) aged 55 or above; (2) self-reported as being diagnosed with type II diabetes mellitus and/or hypertension; (3) able to ambulate independently; (4) able to communicate in Cantonese. The recruited participants were split into two groups: the intervention group (IG) and control group (CG). In the first few batches of recruitment, we split the recruited participants into two groups by random number generation. However, we gradually found that many participants were hesitatant to join the study if the start date of the intervention clashed with their planned events. Therefore, we allowed participants to sign up to the attendance list according to their start-date preference for the six-week intervention. The participants were not aware which start date was for IG and which was for CG. They chose the date that did not clash with the other activities they had already committed to. Waitlist control design was used so that even the participants in the CG could also receive the intervention after completing all the measurements in the study. Participants in the control group received the intervention after the post-intervention assessments at Week 10, that is, from Week 11 onwards. This was a favourable strategy for community-based projects because the participants in the CG did not feel like they were being discriminated against or at a disadvantage. In this study, IG participants were further divided into nine groups with sizes that ranged from nine to 14 participants. IG participants received the six-week intervention during the study period, while CG participants received no intervention and no physical activity information during the study period (Weeks 1–10) but were asked to wear accelerometers at Weeks 0, 6 and 10.

Sample size determination was based on the primary outcome, the number of steps taken per day by an older adult. Since no intervention in the previous studies has incorporated photovoice as one of the components, we could only choose a similar intervention as the reference when we calculated the sample size. We have chosen a cognitive-behavioural intervention which used group discussion to motivate the participants to increase their physical activity level [16]. The target population of this previous study was also similar to our study (patients with type II diabetes), and the effect size of the intervention was small (Cohen’s d = 0.19) [16]. Therefore, we assume the effect size (Cohen’s d) of the intervention in the current study was 0.19, and that a total sample size of 180 (i.e., 90 per group) was enough to detect the between-group difference at 5% significance level and a power of 80% (G*Power 3.0).

### 2.2. Six-Week Intervention

The intervention consisted of six weekly group meetings which involved: (1) the introduction of the concept of photovoice and the importance of doing regular physical activity in chronic illness management; (2) warm-up stretching exercises; (3) capturing photos in the neighbourhood; (4) the sharing of thoughts when the participants reviewed the photos and empowering the participants through reflection (e.g., why they did not do regular physical activity, what made them do physical activity); (5) identifying resources and facilities related to physical activity within the neighbourhood; (6) formulating action plans for physical activity commitment in the next four weeks. The group meetings were arranged for three purposes: (a) building up the participants’ self-efficacy to do exercise; (b) enabling them to identify and review the facilitators and barriers to physical activities through discussion in a group; (c) setting individualised goals in the exercise plan. Each meeting normally lasted for one hour and was arranged under the supervision of a nurse so that the participants could freely exchange their views regarding regular physical activity based on the pre-designed pictorial storybook and their own photos taken around their living environment. A health and fitness officer was also involved in the 3rd meeting (Week 3) to rectify the myths of physical exhaustion and guide the participants to do exercise which prevented unnecessary injury during physical activity. A warm-up stretching exercise was introduced to the participants because many of the older adults were not aware of the importance of warm-up exercises before engaging in physical activity, and this lasted for around 20 min. The purpose of such an introduction was to avoid injury and it was up to the participants to conduct this exercise in their own time. In the 4th and 5th group meetings, the participants discussed the barriers to and facilitators of physical activity (as shown in the photos) and all participants worked together to find possible solutions/strategies to remove barriers for each individual. Group dynamics led to successful problem solving, and participants were empowered to find ways to remove individualised barriers and make use of the facilitators. In the 6th meeting, participants set up individualised action plans (including goals and timetables) for physical activity in the next four weeks according to their preferences and health status. Participants were encouraged to execute the plans during the next four weeks after the meetings.

### 2.3. Outcomes and Assessment

The primary outcome was the average number of steps taken daily. Participants were invited to wear accelerometers at baseline (Week 0), right after the six-week intervention (Week 6), and four weeks after the intervention (Week 10). In the previous accelerometer-based interventional study for older adults with type II diabetes [16], participants were invited to wear accelerometers for five consecutive days. We followed this arrangement; therefore, in the current study, each participant was invited to wear an accelerometer for five days (24 h per day) in each specific period. A very recent study also showed that a minimum of five consecutive days of accelerometer monitoring could ensure reliability in estimating sedentary behaviour and measuring physical activity among the older adults [17]. In this study, accelerometers were distributed to the older adults on Day 1 (this could be in the middle of the day) and were returned on Day 7; we could obtain five complete days after trimming off the day of distribution and day of return. As the trimmed off days were weekdays, the composition of weekdays and weekends in these five days were the same for each participant. The data was automatically stored in a highly secured server. The trained research assistant retrieved the data by using a password. The average number of steps taken daily was calculated by dividing the total number of steps taken in the five days by five.

There were three secondary outcomes:

(1) Senior fitness tests [18] consisting of seven tests: (i) a six-minute walk test measures the distance walked within six minutes to reflect aerobic endurance. The higher the value, the better the aerobic endurance; (ii) a 30-s chair stand test measures the number of stands that could be made from a sit-down position in 30 s and this reflects lower body strength. The higher the value, the better the lower body strength; (iii) the eight-feet up-and-go test measures the time required (in seconds) to stand up and travel an eight-foot distance from an initial sitting position, and this also reflects agility and dynamic balance. The lower the value, the better the agility and balance; (iv) the arm curl test measures the number of repetitions that a dumbbell can be lifted in 30 s (a 5-lb. weight for females and an 8-lb. weight for males), and this reflects upper body strength. The higher the value, the better the upper body strength; (v) a back scratch test measures the distance (in cm) reached by the two hands when they are extended to scratch the back, and this reflects upper body flexibility. The higher the value, the better the upper body flexibility; (vi) the handgrip test measures the strength of the hand, and it reflects upper body strength. The higher the value, the better the grip strength; vii) the chair sit-and-reach test measures the distance reached by the hands to the toes when sitting on a chair, and this reflects lower body flexibility. The higher the value, the better the lower limb flexibility.

(2) Self-efficacy for doing physical activity was measured by the nine-item Self-efficacy Scale for Exercise–Chinese version (SEE-C) [19]. Items of the scale were rated on an 11-point Likert scale from 0 (not confident) to 10 (very confident) and were summed to generate a total SEE-C score, with higher scores indicating more confidence in performing the exercise. Good psychometric properties of SEE-C were reported in a local study [20]. The internal consistency of SEE-C, measured by Cronbach’s alpha, was 0.75 [19].

(3) Barriers to exercise were measured by the 23-item Chinese Barriers to Exercise Scale (CBES) [21]. Items of the scale were rated on a five-point Likert scale from 1 (very disagree) to 5 (very agree) and are summed to generate a total CBES score, with higher scores indicating more barriers to performing exercise. Good psychometric properties of CBES were reported in a local study [21]. The internal consistency of CBES in the current study, measured by Cronbach’s alpha, was 0.69 [21].

These were measured by trained research assistants in the health assessments at Week 0 and Week 10. Demographics which included sex, age, marital status, educational level, and employment status, districts in which they lived in, living status, health literacy (measured by the Chinese Health Literacy Scale for Chronic Care (CHLCC) [22]) and types of chronic illnesses were collected. The internal consistency of CHLCC, measured by Cronbach’s alpha, was 0.91 [22].

Blinding: In this study, outcome assessors were blind to the participant’s group allocation. However, the participants could not take part blindly because the participants knew whether they had undergone the six-week intervention or not. To minimise bias, the participants in the CG were not told that the second assessments were considered as post-intervention assessments.

### 2.4. Ethical Statement

All participants gave their informed written consent for inclusion before they participated in the study. The study was conducted in accordance with the Declaration of Helsinki, and the protocol was approved by the Institutional Review Board of the University of Hong Kong/Hospital Authority Hong Kong West Cluster (HKU/HA HKW IRB) (reference number: UW14-447).

### 2.5. Data Analysis

Descriptive statistics, including numbers, percentages, means and standard deviation (SD) for normally distributed variables, and medians and inter-quartile range (IQR) for non-normally distributed variables, were performed on demographics, the number of steps taken daily, and self-efficacy for and barriers to doing exercise. Chi-square tests for categorical variables, an independent t-test for normally distributed continuous variables, and a Mann–Whitney test for non-normally distributed continuous variables were used to compare the similarity in baseline characteristics between the two groups. The Kolomogrov–Smirnov Test was used to check the normality assumption of the variables. Generalised estimating equations (GEE) models were used to assess the intervention effect over time on all the outcome variables, including the number of steps taken daily, physical fitness variables, self-efficacy for and barriers to performing physical activities. In the GEE models, Time, Group and the interaction term between Time and Group (Time X Group) are independent variables. The coefficient of the interaction term Time X Group estimates the mean difference in the change of the outcome variable over time between the two treatment groups. A significant result for the Time X Group indicates a significant differential change in the outcome variable over time between the two groups. Since the participants’ age was found to be a confounding variable, age was controlled in the GEE models for all the outcome variables.

## 3. Results

### 3.1. Participant Recruitment and Retention

A total of 252 participants were screened for eligibility. Among the 225 eligible participants, 204 consented to join the study, with 107 participants assigned to IG and 97 to CG. For accelerometer assessment, the attrition rates in IG were 19% at both Week 6 and Week 10 while those for CG were 22% and 24%, respectively, with no statistically significant differences found between the two groups (Week 6: *p* = 0.603; Week 10: *p* = 0.384). For secondary outcomes which were followed-up only at Week 10, there was no statistically significant difference in the attrition rates for IG and CG (25% versus 31%, *p* = 0.363).

### 3.2. Descriptive Statistics

Table 1 showed the demographics of the participants in this project. The mean age of the participants was 73.3 (SD = 7.5). The majority of the participants were females (75.0%), married (54.9%), had primary education or below (53.5%) and retired (83.3%). The mean score of health literacy was 38.7 (SD = 9.8). There was no statistically significant difference in demographics and the health literacy level between IG and CG.

### 3.3. Evaluation of the Make a Change through Photovoice (MCPv) Programme

#### 3.3.1. Average Number of Steps Taken Daily

The average number of steps taken per day was measured at three time points (Week 0, Week 6, and Week 10). A Kolomogrov Smirnov Test showed that the normality assumption of the variable was not rejected (*p* = 0.056). As shown in Table 2, the average number of steps taken by the IG participants increased at Week 6 after intervention and then decreased at Week 10 to a level similar to that of Week 0 while that of the CG participants decreased gradually from Week 0 to Week 10.

A statistically significant difference in the changes in the average number of steps taken daily between the two groups at Week 10 was observed (estimated group mean difference = 965.4, 95% confidence interval, CI 92.2, 1838.6, *p* = 0.030), as shown in Table 2.

#### 3.3.2. Physical Fitness

A Kolomogrov–Smirnov Test showed that the normality assumption in all fitness test items was rejected with *p* < 0.05, except in the six-minute walk test and back scratch test. The results in Table 3 revealed that there were statistically significant differences in the changes in the 30-s chair stand and chair sit-and-reach tests from Week 0 to Week 10 between IG and CG. For the 30-s chair stand test, both IG and CG groups showed improvement over time but the increment in IG was significantly greater than the CG (*p* = 0.043). For the chair sit-and-reach test, IG participants showed improvement from −2.89 to −1.28 while CG participants showed worsening from −1.22 to −1.65 over time. Participants in both groups showed improvements in the six-minute walk test, arm curl test, and back scratch test but reductions in the eight-feet up-and-go test and handgrip test; however, no statistically significant differences between the groups were observed.

#### 3.3.3. Health-Related Variables

A Kolomogrov–Smirnov Test showed that the normality assumption of the SEE-C and barriers to doing exercise were rejected with *p* < 0.05. There was no statistically significant difference in the mean score difference of SEE-C between IG and CG from Week 0 to Week 10, although IG participants showed a slow decline in their SEE-C level compared to the CG participants (Table 4). For barriers to performing physical activities, there was also no statistically significant between-group difference, but the decrease in the barrier level in IG was greater than that in the CG (Table 4).

## 4. Discussion

This study provided evidence of the effects of this intervention on the physical activity level and physical fitness among patients with diabetes and/or hypertension. Photo taking, photo sharing in group discussion, identifying facilitators and barriers to exercise, self-reflection, and action plans formed a comprehensive strategy to encourage older adults with chronic illnesses to engage in physical activities. The IG participants had a higher average number of steps taken per day than the CG participants at Week 10, and the IG participants’ lower body strength and flexibility showed greater improvement than the CG participants. These results implied that the multiple components of the six-week intervention not only motivated the participants to consider behavioural changes but also take action. The primary outcome (number of steps per day) was measured by accelerometers; however, the use of accelerometers was evidenced to motivate older adults to increase their physical activity levels [23]. In the current study, we isolated the effect of accelerometers by asking the participants to use the accelerometers before and after the intervention, and these were used by both the intervention group and the control group. The accelerometer effect was, therefore, balanced out due to this design. Thus, the change in physical activity was not related to the accelerometer effect. We also noticed that both IG and CG participants’ average number of steps taken per day at baseline was very high, this may partially explain why only minor changes were observed after intervention at Week 10.

In the current study, photos serve as a means for initiating the group discussions. The photovoice process has been integrated into the intervention, converging with other components of the intervention. Incorporating photovoice as a kind of intervention was a new strategy. Woda et al. in 2018 [9] also advocated this new approach. They found that photovoice provided the opportunity for elderly African Americans to share their beliefs and perspectives with regard to self-care, and this easy-to-use intervention eventually empowered vulnerable participants [9]. Another study had also tried this new approach and considered photovoice as an intervention, measuring the outcome (that is, quality of life) after the participants went through the photovoice process [24]. The finding of the current study provided additional evidence that photovoice can be extended from a research methodology to an intervention, and that this was an appropriate intervention for older adults.

Self-reflection of current health status is a crucial strategy in diabetes management. As shown in another intervention that motivated diabetic patients to conduct self-reflection upon getting readings from continuous glucose monitoring devices, the patients committed to changing their behaviour; especially, exercise behaviour and diet control [25]. The finding of the current study extended our understanding of self-reflection. Self-reflection can be supported by a group of people; the group mates discussed the barriers that only some individuals may face and then made suggestions on how to remove these barriers. This kind of support would be helpful to those individuals who had never thought of or worked out any solutions to their existing problems.

Another important strategy in the current intervention was action planning. Participants were invited to develop a plan for exercise by reviewing their own context (including the environment, the barriers that they could remove eventually, and the facilitators that they would use). Action planning was considered as an essential component in diabetes self-management programmes [26,27]. A recent systematic review reported that action planning was one of the most commonly used techniques in physical activity interventions amongst older adults [28]. Unlike the study undertaken by O’Donnell et al. in 2018 [29], which asked the participants to self-report whether they conducted the actions at 1 week and 3 months after planning the action (doing exercise), the current study conducted subjective measurements (using accelerometers and senior fitness tests) to assess the actions taken at Week 10 (which was four weeks after planning the action).

Unexpectedly, there was no significant change in self-efficacy to exercise (SEE-C) or significant reduction of the barriers to physical activities (CBES). Since SEE-C was only measured twice, in Week 0 and Week 10, we suspect that SEE-C may follow a similar trajectory to that of the number of steps taken (i.e., increased from Week 0 to Week 6, but dropped back down during Week 10) and, therefore, the MCPv programme might have a positive effect on SEE-C. However, the effect faded as time passed—in such a way that our data collection might not have been able to detect the effect. This finding mirrored the findings of the previous study in which self-efficacy to exercise could be improved by intervention, but it had a short-term effect because older adults often overestimate their self-efficacy to exercise, and when they actually participate in physical activity, their self-efficacy decreased [30]. Further research is warranted to investigate when and how self-efficacy decreases.

Other than the key findings of the study, we made some additional observations. To some participants, digital cameras were another kind of new technology. The challenges were: how to retrieve the photos, how to download the photos onto the computer and how to project the photos onto the screen for sharing. The Project Coordinator played an important role in supporting these activities. The Project Coordinator talked to each participant, discussed which photos should be taken for sharing, and prepared all photos prior to the group meetings. This alleviated the participants’ stress to share their photos within the group.

Through the discussion with the photos, we further understand the views of the older adults regarding the relationship between the environment and their engagement in physical activities. A few photos taken by the participants indicated that road renovation or a nearby construction site was one of the barriers to physical activity because they could not get access to the park to perform physical activities due to roadblocks. Such comments echoed the finding of the previous study in which walkability and physical exercise were associated but there was no interaction between the two variables, that is, a high level of walkability in the neighbourhood was associated with a high level of physical exercise [31]. In view of this, the government should develop strategies to minimise the duration of roadblocks and develop alternate pathways during renovation periods so that the environment can maintain its walkability and older adults can still engage in physical activity (mainly walking) in the neighbourhood.

This study had several limitations. Firstly, there was a high potential for measurement error as the measurements of senior fitness tests were taken by more than one person. Although we trained all outcome assessors, the chance of obtaining errors in these measurements was still high. Caution should be taken when interpreting these data. Secondly, the samples were recruited from the centres for the elderly and, therefore, this sample may not represent all community-dwelling older adults. Not all the older adults would join the membership of the centres for the elderly, and this may have led to selection bias. Thirdly, the findings may have a certain level of bias because it could not be a blind-participation study as the participants would know they had received the intervention or not before the second assessment. Moreover, the current findings may be subject to selection bias because we could not allocate the participants to the intervention or the control group randomly due to practical considerations, although we did not find a statistically significant difference in baseline characteristics between the two groups. Lastly, some data in the accelerometers could not be collected because participants refused to wear the accelerometers due to skin irritation and feelings of discomfort.

## 5. Conclusions

The MCPv programme successfully increased participants’ physical activity level and physical fitness, particularly their lower body strength and lower body flexibility. The protocol of this intervention could further support the development of other multi-component interventions in other health areas in the Chinese community. The study provided evidence to health professionals that older adults were capable of expressing their views regarding the barriers to and facilitators of physical activities, reflecting their behaviour and changing it as necessary. Knowing these stakeholders’ responses to the intervention, health professionals could develop relevant health promotional activities for this population.

## Figures and Tables

**Table 1 ijerph-16-01219-t001:** Participants’ demographics.

Demographics	Intervention (*n* = 107)		Control (*n* = 97)		Total (*n* = 204)		
	Count (%)	Mean ± SD	Count (%)	Mean ± SD	Count (%)	Mean ± SD	*p*
Age		73.6 ± 7.5		73.1 ± 7.5		73.3 ± 7.5	0.720
Sex							0.453
Female	79 (73.8%)		74 (76.3%)		153 (75.0%)		
Male	28 (26.2%)		23 (23.7%)		51 (25.0%)		
Marital status							0.732
Single	7 (6.5%)		8 (8.2%)		15 (7.4%)		
Married	59 (55.1%)		53 (54.6%)		112 (54.9%)		
Others	41 (38.4%)		36 (37.2%)		77 (37.7%)		
Education level							0.814
No formal education	24 (22.4%)		19 (19.6%)		43 (21.1%)		
Primary	34 (31.8%)		32 (33.0%)		66 (32.4%)		
Secondary	37 (34.6%)		33 (34.0%)		70 (34.3%)		
Tertiary	10 (9.3%)		13 (13.4%)		23 (11.3%)		
Missing	2 (1.9%)		0 (0%)		2 (1.0%)		
Employment status							0.780
Retired	88 (82.2%)		82 (84.5%)		170 (83.3%)		
Others	19 (17.8%)		15 (15.5%)		34 (16.7%)		
Districts							0.444
Western	14 (13.1%)		12 (12.4%)		47 (20.9%)		
Others	93 (86.9%)		85 (87.6%)		178 (79.1%)		
Living status							0.844
Living Alone	33 (30.8%)		30 (30.9%)		63 (28.0%)		
Living with Spouses	25 (23.4%)		30 (30.9%)		55 (24.4%)		
Living with Children	21 (19.6%)		14 (14.4%)		35 (15.6%)		
Others	28 (26.2%)		23 (23.8%)		51 (25.0%)		
Health literacy		37.8 ± 10.5		39.7 ± 9.1		38.7 ± 9.8	0.190
Chronic illness							
Hypertension	71 (73.2%)		68 (70.1%)		139 (61.8%)		0.781
Diabetes	31 (32.0%)		23 (23.7%)		54 (24%)		0.755

**Table 2 ijerph-16-01219-t002:** Generalised estimating equations (GEE) results on the average number of steps taken daily over time between the intervention group (IG) and control group (CG).

Time for Measuring	Intervention (*n* = 107)	Control (*n* = 97)	GEE results	
	Mean ± SD	Mean ± SD	Estimated Group Mean Difference (95% Confidence Interval)	*p*
Week 0	11,506 ± 4428	12,522 ± 4624		
Week 6	12,211 ± 4681	12,316 ± 5013	777.6 (–35.3, 1590.5)	0.061
Week 10	11,686 ± 4594	11,643 ± 4803	965.4 (92.2, 1838.6)	0.030

*Note.* Adjusted for age.

**Table 3 ijerph-16-01219-t003:** GEE results of fitness test between IG and CG.

Variables of Senior Fitness Test	Intervention (*n* = 107)	Control (*n* = 97)	GEE result	
	Median (IQR)	Median (IQR)	Estimated Group Mean Difference (95% Confidence Interval)	*p*
6-min Walk Test				
Week 0	426.00 (147.00)	432.00 (102.00)		
Week 10	426.00 (124.50)	444.00 (111.00)	−4.010 (−17.77, 9.75)	0.568
30-s Chair Stand Test				
Week 0	12.00 (6.00)	12.00 (7.00)		
Week 10	13.00 (5.00)	13.00 (7.00)	0.967 (0.029, 1.904)	0.043
8-feet Up-and-Go Test				
Week 0	6.43 (2.57)	6.21 (2.62)		
Week 10	6.26 (2.56)	6.03 (2.46)	−0.179 (−0.607, 0.249)	0.413
Arm Curl Test				
Week 0	14.00 (5.00)	14.00 (5.75)		
Week 10	14.50 (5.00)	14.50 (5.00)	−0.344 (−1.261, 0.574)	0.463
Back Scratch Test				
Week 0	−11.10 (-15.86)	−8.28 (16.91)		
Week 10	−6.25 (19.38)	−5.50 (18.29)	0.985 (−1.023, 2.994)	0.336
Handgrip Test				
Week 0	19.13 (7.09)	19.15 (7.46)		
Week 10	18.74 (8.11)	18.16 (7.58)	0.500 (−1.284, 2.284)	0.583
Chair Sit-and-Reach Test				
Week 0	−1.50 (11)	0.88 (16.44)		
Week 10	0.00 (12.25)	0.88 (16.31)	2.068 (0.404, 3.731)	0.015

*Note.* Models were adjusted for age. IQR = inter-quartile range.

**Table 4 ijerph-16-01219-t004:** GEE results of fitness test items between IG and CG.

Measures	Intervention (*n* = 107)	Control (*n* = 97)	GEE results	
	Median (IQR)	Median (IQR)	Estimated Group Mean Difference (95% Confidence Interval)	*p*
Self-efficacy for doing physical activities				
Week 0	5.56 (3.00)	5.78 (3.00)		
Week 10	5.22 (2.89)	5.44 (2.78)	0.303 (−0.137, 0.743)	0.177
Barriers to doing physical activities				
Week 0	2.26 (1.11)	2.09 (0.87)		
Week 10	2.09 (0.87)	2.13 (0.95)	−0.049 (−0.285, 0.187)	0.683

*Note.* Models were adjusted for age. IQR = inter-quartile range.

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
