# Peer review of "Motivating Diabetic and Hypertensive Patients to Engage in Regular Physical Activity: A Multi-Component Intervention Derived from the Concept of Photovoice"

_ijerph, 2019, doi:10.3390/ijerph16071219_

Round 1

Reviewer 1 Report

The authors sufficiently addressed the comments from previous round. I have no further comments.

Author Response

Thank you

Reviewer 2 Report

This is an interesting manuscript yet I have significant concerns about the study methodology and outcomes. The authors have stated this is a quasi-experimental design yet it actually looks like a sequential mixed methods study with the first part being photo voice to establish barriers to physical activity then the intervention which is the later groups sessions. Photo voice is not an intervention as you stated in the paper, it is a qualitative methodology therefore should not be used to intervene but instead should be used to either inform or evaluate an intervention. Setting up the study in the way that you have means that your primary outcomes do not match your intervention. The only outcome feasible if photo voice is the intervention is the one you stated in your conclusion - "older adults were capable to express their views about the barriers and facilitators to physical activities". This conclusion firstly does not reflect your abstract conclusion and notably is not known to affect physical activity levels.

It may be better to describe photo voice as being a component of the group sessions instead of saying it is the intervention given these points, though really it is the exploratory part of the project

Further comments:

Introduction: 

Line 72, the authors should describe why physical activity at the recommended level is important for disease management

Multiple grammatical errors highlighted in the attached manuscript

Methods 

Line 87 "CBPR" needs defining

Line 90 - this is not a quasi experimental study. If randomisation was done then it is a randomised controlled trial (lacking blinding etc)

Lines 93 +94 - what do the authors mean by batches of recruitment? Needs further explanation

Circa line 98 How was T2DM and hypertension identified - by self report? If these are not  outcomes of the study, why is it being measured? Id it is contextual information then diabetes and hypertension should be discussed more fully in the background (and why PA is important in the prevention and management of the conditions

Line 105, how were participants divided into groups (on what basis) needs clarifying further

Line 111 - not sure how the sample size can be calculated using photo voice, again I urge the authors to think about their description of the intervention

Circa line 115 Is building self-efficacy/setting exercise plans not your intervention? I am struggling to see how photovoice changes anything - I believe this is reflected in the non-significance difference in steps at week 6. The title here is also misleading as it sounds as though one intervention at 6 weeks was the intervention

Paragraph from line 148 onwards, The 6 minute walk text needs a corresponding citation (as well as any other tests that appear in the literature

Line 176 How was health literacy measured i.e. which scale was used?

Line 191 - why is the arm curl text not included in the GEE analysis - an explanation is needed

Multiple grammatical errors highlighted in the attached manuscript

Discussion

Multiple grammatical errors highlighted in the attached manuscript

The whole discussion is i need of improvement as it generally lacks relevance to the actual study findings as well as any comparison to the wider literature or practice recommendations. For much of the discussion, the secondary aims discussed are not found in the results section nor listed as study outcomes. This section starts with the authors stating there are 7 main findings before they go on to list these 'findings' in order'. I would suggest that the authors instead focus on the outcomes they set out to assess then link these into these other incidental findings. In addition:

Line 270 - stating that photo voice is an effective health promotion strategy to  improve physical fitness of patients with diabetes or hypertension is quite and overstatement as first, there is mention of verification of diabetes or hypertension status and second, there are a number of methodological limitations to this study that affect this conclusion (no blinding, high potential for measurement error and low representativeness of the sample, none of which are discussed in the limitations. More use of hedging language here would be useful.

Lines 278 - circa 282 The authors indicate that self-efficacy is not sustained and recommend with continuing sessions. How does this affect sustainability fo the intervention? There needs to be much more discussion around this as in realty, continued sessions are unlikely to be feasible unless delivered by a volunteer.

Line 282 -circa 286 The authors state that "The instructors of the programme should observe participants’ pain level during and after the programme, and monitor the duration of the programme to avoid the participants from exhaustion. With this monitoring, older adults with chronic illness would enjoy physical activities and be confident to participate in the activities." Again I find this to be an overstatement and not related to the stated outcomes of the study or study results. This is not a finding but an observation of the authors and should be described as such

Line 287 'forth'  (which should read fourth) the study has some unique features. Again this is not a finding but an observation of the authors and should be described as such. Further that whole paragraph is confusingly written and needs to be shortened and clarified if it is to be included

Line 302 'fifth' (I discourage numbering anything like this, even true study findings), again not a finding but an author observation (older adults and technology)

Line 312 'Sixth' This is not a finding but instead related to measurement (and therefore should be with other comments in study validity)

Line 322 'Further investigation of the impact of environmental factors 322 on physical activities is warrant.' There is a significant body of literature on this issue. If the authors are referring to older people in China, they should state as much

Limitations paragraph:

The authors say that the exercise intervention was not vigorous enough. First, this sounds appropriate for older adults and in any case - the intervention was supposedly the photo voice so this should not make any difference (as wasn't the idea that the older adults set their own exercise agenda)? 

If blood pressure was monitored there should be a full description of how this was done in the methods section (as well as ascertainment of diabetes status)

A number of significant limitations have been overlooked including potential for measurement error, selection bias, lack of blinding and lack of protocol which may have reduced efficacy the intervention itself

Author Response

Comments and Suggestions for Authors
R2(Comment 1): This is an interesting manuscript yet I have significant concerns about the study methodology and outcomes. The authors have stated this is a quasiexperimental design yet it actually looks like a sequential mixed methods study with the first part being photo voice to establish barriers to physical activity then the intervention which is the later groups sessions.
Our Response : Sorry for the confusion made. We should make clear that we have developed the 6-week intervention with photovoice as part of the components and we are using quasi-experimental design to evaluate the efficacy of this 6-week intervention. Randomization of samples was not performed. We added the following sentences to Abstract (line 17-21): Aims: A community-based intervention with photovoice as part of the components was developed to promote physical activity (PA) among patients with diabetes and hypertension. This study aimed to evaluate the efficacy of this intervention on health behaviour (walking) and health outcomes. Design: A non-randomized controlled trial…
We have kept “This was a non-randomized controlled trial…” (line 109) in the Section 2.1 Design and participants.
R2(Comment 2): Photo voice is not an intervention as you stated in the paper, it is a qualitative methodology therefore should not be used to intervene but instead should be used to either inform or evaluate an intervention. Setting up the study in the way that you have means that your primary outcomes do not match your intervention. The only outcome feasible if photo voice is the intervention is the one you stated in your conclusion - "older adults were capable to express their views about the barriers and facilitators to physical activities". This conclusion firstly does not reflect your abstract conclusion and notably is not known to affect physical activity levels. It may be better to describe photo voice as being a component of the group sessions instead of saying it is the intervention given these points, though really it is the exploratory part of the project.
Our Response : Thank you for your comments and suggestions. This is the innovation of the study. We acknowledge photovoice as a powerful tool to empower individuals (the participants) and it was used a qualitative method. However, we also think that through the process of discussion in groups and using the photos, the participants

are actually experiencing personal growth. So we add the following sentences in Introduction (line 68-72): … we acknowledge photovoice as a powerful tool to empower individuals by giving them opportunities to talk about their individual problems (for example, having certain barriers to physical activities) and find ways to solve the problems. Through this process, empowerment occurred and this would support individual growth (for example, participants may consider to make a change in behavior).
Introduction (Line 75-87): In the previous studies, photovoice was considered as a qualitative method to collect participants’ views on a health-related topic and eventually their comments help generate policy recommendations. However, no study has ever investigated the actual effect of this kind of photo-oriented group discussion on health behaviors and the possible consequences on the individuals per se (changes in health outcomes after behavioral change). We therefore deliberately formulated a 6-week community-based health promotion programme (or the intervention) with photovoice as part of the components in the intervention. This intervention did not solely use photos and group discussion, it also comprised of other components such as understanding the importance of physical activity in chronic illness management through reading the comic books, developing action plans for the next 4 weeks in the neighbourhood. The innovation of this study is the incorporation of the concept of photovoice in an intervention as we believe that the participants would have personal growth and make decision on their behaviour after their participation in the intervention.
We removed the following sentences in Introduction (line 73-75):
Nonetheless, these previous studies were qualitative studies, which showed the process of the use of photos in behavioural change and its impacts on psychosocial health. Little is known about the effect of photovoice on physical health and actual behaviour in daily lives over time.
Further comments:
Introduction:
R2(Comment 3): Line 72, the authors should describe why physical activity at the recommended level is important for disease management.
Our Response : We have added a few sentences to explain why physical activity at recommended level is important in diabetes and hypertension management (line 88-92):
For older adults with type 2 diabetes and/or hypertension, maintaining physical activity at a recommended level (for example, 30 minutes of moderate exercise 5 times per week or 150 minutes per week) [13] is of high priority in disease management. We target for this population because physical activity at the recommended level plays an important role in glycemic control for type 2 diabetes patient [13] and helps maintain normal blood pressure for hypertensive patients [14].

Methods:
R2(Comment 4): Line 87, "CBPR" needs defining.
Our Response : Since we have stated that photovoice is part of the components of the intervention, mentioning CBPR and social ecological model of health may lead to confusion. To avoid this, we have deleted the description of CBPR and social ecological model of health (line 45-53; line 96; line 106-108)
R2(Comment 5): Line 90 - this is not a quasi experimental study. If randomisation was done then it is a randomised controlled trial. (lacking blinding etc)
Our Response : It is a non-randomised controlled trial, and we have deleted the word ‘by random number generations’. We cannot enforce straight randomization in this community-based programme as the collaborator objected about the arrangement. We therefore allowed the participants to choose the start date of the 6-week intervention without knowing which one the intervention group is and which one the control group is. This is the best arrangement we could make in the community.
R2(Comment 6): Lines 93 +94 - what do the authors mean by batches of recruitment? Needs further explanation
Our Response : We have added the explanation (line 113-115): For each batch of recruitment, we recruited 9-12 participants only. Such arrangement was made because we had limited number of digital cameras that could be loaned to the participants.
R2(Comment 7): Circa line 98 How was T2DM and hypertension identified - by self report? If these are not outcomes of the study, why is it being measured? If it is contextual information then diabetes and hypertension should be discussed more fully in the background (and why PA is important in the prevention and management of the conditions)
Our Response : Yes, T2DM and hypertension were self-reported, and these are for screening the potential participants to join the intervention. We gave reasons why regular physical activity in persons with hypertension and/or diabetes in Introduction. We have added the following sentences (line 90-92): We target for this population because physical activity at the recommended level plays an important role in glycemic control for type 2 diabetes patient [13] and helps maintain normal blood pressure for hypertensive patients [14].
In 2.1 Designs and participants (line 118-119), we added: self-reported as being diagnosed with type 2 diabetes mellitus and/or hypertension
R2(Comment 8): Lines 105, how were participants divided into groups (on what basis) needs clarifying further.
Our Response : We have added the following sentences to indicate how the participants were divided into groups (line 121-124):

Participants were allowed to sign up in the attendance list according to their preference of the start date of the 6-week intervention. The participants were not be aware that which start date was for IG and which was for CG. They chose the date that did not crash with the other activities they committed.
R2(Comment 9): Lines 111, not sure how the sample size can be calculated using photo voice, again I urge the authors to think about their description of the intervention.
Our Response : The sentence was modified as: (In line 135-142) Since no intervention in the previous studies has incorporated photovoice as part of the components, we could only choose a similar intervention as the reference when we calculated the sample size. We have chosen a cognitive-behavioural intervention which used group discussion to motivate the participants to increase their physical activity level [16]. The target population of this previous study was also similar to ours (patients with type 2 diabetes) and the effect size of the intervention was small (Cohen’s d = 0.19) [16].
R2(Comment 9): Circa line 115 Is building self-efficacy/setting exercise plans not your intervention? I am struggling to see how photovoice changes anything – I believe this is reflected in the non-significance difference in steps at week 6. The title here is also misleading as it sounds as though one intervention at 6 weeks was the intervention???
Our Response : The intervention, with photovoice as part of the components, aims to build up individuals’ self-efficacy to do exercise and set up their own action plans for 4 weeks (from Week 7 to 10). This explained why Week 6 did not have a favourable outcomes (increase in physical activity level). We elaborated the description of the 6-week intervention and the added parts are in YELLOW (line 146-151):
The intervention was six weekly group meetings which involved: 1) the introduction of the concept of photovoice and the importance of doing regular physical activity in chronic illness management, 2) warm-up stretching exercise, 3) capturing photos in the neighborhood, 4) sharing the thoughts when the participants reviewed the photos and empowering the participants through reflection (e.g. why they did not do regular physical activity, what made them do physical activity), 5) identifying resources and facilities related to physical activity within the neighborhood, and 6) formulating action plans for physical activity commitment.
We propose to change the title of the paper to: Motivating Diabetic and Hypertensive Patients to Engage to Regular Physical Activity: A Multi-component Intervention with The Use of Photos, Group Discussion and Action Planning
R2(Comment 10): Paragraph from line 148 onwards, The 6 minute walk text needs a corresponding citation (as well as any other tests that appear in the literature
Our Response : All the 7 tests (6-minute walk test, 30-second chair stand test, 8-feet up-and-go, etc.) are the components of Senior Fitness Test. Reference [18] was added (line 188).
R2(Comment 11): Line 176 How was health literacy measured i.e. which scale was used?

Our Response : Health literacy was measured by the validated scale “Chinese Health Literacy for Chronic Care (CHLCC)”. Reference and psychometric properties were added (line 218-220): …health literacy (measured by Chinese Health Literacy Scale for Chronic Care (CHLCC) [22]) and types of chronic illnesses were collected. The internal consistency of CHLCC measured by Cronbach’s alpha, is 0.91 [22].
R2(Comment 12): Line 191 - why is the arm curl text not included in the GEE analysis – an explanation is needed
Our Response : Sorry for the confusion made. We have double checked the GEE analysis in Table 3. We have controlled ‘age’ in ‘arm curl test’ as well. We have removed the following part:
…. Age was controlled in the GEE models for all the outcome variables except Arm Curl Test. (line 246) The note under Table 3 (line 300): Models were adjusted for age.
Discussion:
R2(Comment 13): The whole discussion is in need of improvement as it generally lacks relevance to the actual study findings as well as any comparison to the wider literature or practice recommendations. For much of the discussion, the secondary aims discussed are not found in the results section nor listed as study outcomes. This section starts with the authors stating there are 7 main findings before they go on to list these 'findings' in order'. I would suggest that the authors instead focus on the outcomes they set out to assess then link these into these other incidental findings.
Our Response : As photovoice has been stated as part of the components of the intervention, we do not highlight the participants’ opportunities to express their views through photos. The discussion was rewritten addressing the findings and its implications on practice.
R2(Comment 14): Line 270 - stating that photo voice is an effective health promotion strategy to improve physical fitness of patients with diabetes or hypertension is quite and overstatement as first, there is mention of verification of diabetes or hypertension status and second, there are a number of methodological limitations to this study that affect this conclusion (no blinding, high potential for measurement error and low representativeness of the sample, none of which are discussed in the limitations. More use of hedging language here would be useful.
Our Response : The sentence ‘photovoice is an effective health promotion strategy ….’ was deleted.
The following sentences were added to Method to explain the blinding (line 221-224): In this study, data collectors were blinded as the data collectors did not know who were in IG and who are in CG. However, the participants could not be blinded

because the participants knew whether they had undergone the 6-week intervention or not. To minimize bias, the participants in the CG were not told that the second assessments were considered as post-intervention assessments.
The limitations were added (line 426-437): This study has several limitations. Firstly, there was a high potential for measurement error as the measurements of senior fitness tests were taken by more than one person. Although we trained all outcome assessors, the chance of getting error in these measures was still high. Caution should be taken when interpreting these data. Secondly, the samples were recruited from the elderly centres and therefore this sample may not represent all the community-dwelling older adults. Not all the older adults would join the membership of the elderly centres, and this may lead to selection bias. Thirdly, the findings may have certain level of bias because blinding could not be made to the participants as they would know they had received the intervention or not before the second assessments. Moreover, the current findings may subject to selection bias because we could not allocated the participants to the intervention or the control group randomly due to practical consideration although we did not find significant difference in baseline characteristics between the two groups.
R2(Comment 15): Lines 278 - circa 282 The authors indicate that self-efficacy is not sustained and recommend with continuing sessions. How does this affect sustainability for the intervention? There needs to be much more discussion around this as in realty, continued sessions are unlikely to be feasible unless delivered by a volunteer.
Our Response : We have deleted this sentence. Discussion about the change of self-efficacy over time was added: (line 377-381) This finding mirrored the findings of the previous study in which self-efficacy to exercise could be improved by intervention, but it had short-term effect because older adults often overestimate their self-efficacy to exercise, but when they actually participate in physical activity, their self-efficacy would decrease (Olson & McAuley, 2015). Further research is warrant to investigate when and by what self-efficacy decrease.
R2(Comment 16): Line 282 -circa 286 The authors state that "The instructors of the programme should observe participants’ pain level during and after the programme, and monitor the duration of the programme to avoid the participants from exhaustion. With this monitoring, older adults with chronic illness would enjoy physical activities and be confident to participate in the activities." Again I find this to be an overstatement and not related to the stated outcomes of the study or study results. This is not a finding but an observation of the authors and should be described as such
Our Response : Agree. We deleted these sentences.

R2(Comment 17): Line 287 'forth' (which should read fourth) the study has some unique features. Again this is not a finding but an observation of the authors and should be described as such. Further that whole paragraph is confusingly written and needs to be shortened and clarified if it is to be included
Our Response : This paragraph was deleted.
R2(Comment 18): Line 302 'fifth' (I discourage numbering anything like this, even true study findings), again not a finding but an author observation (older adults and technology)
Our Response : Since not many literatures reported how to facilitate older adults to participate in an intervention that included photo-taking and photo-sharing, we think it is worthy to share our observations. We shortened this paragraph and indicated clearly that these are our observations.
R2(Comment 19): Line 312 'Sixth' This is not a finding but instead related to measurement (and therefore should be with other comments in study validity)
Our Response : This paragraph was moved to line 327-332.
R2(Comment 20): Line 322 'Further investigation of the impact of environmental factors on physical activities is warrant.' There is a significant body of literature on this issue. If the authors are referring to older people in China, they should state as much
Our Response : We deleted that sentence and added some discussion about the environment, walkability and older adults’ physical activity: (line 417-423) Such comments echoed the finding of the previous study in which walkability and physical exercise were associated but there was no interaction between the two variables, that is, high level of walkability in the neighbourhood was associated with high level of physical exercise (Jia et al., 2018). In view of this, government should develop strategies to minimize the duration of road block and develop alternate pathway during renovation period so that the environment could maintain its walkability and older adults could still engage in physical activity (mainly walking) in the neighborhood.
Limitations paragraph:
R2(Comment 21): The authors say that the exercise intervention was not vigorous enough. First, this sounds appropriate for older adults and in any case - the intervention was supposedly the photo voice so this should not make any difference (as wasn't the idea that the older adults set their own exercise agenda)?
Our Response : We deleted these sentences to avoid confusion.

R2(Comment 22): If blood pressure was monitored there should be a full description of how this was done in the methods section (as well as ascertainment of diabetes status)
Our Response : Blood pressure and glycemic control were not the outcomes of the study. The sentences related to these were deleted.
R2(Comment 23): A number of significant limitations have been overlooked including potential for measurement error, selection bias, lack of blinding and lack of protocol which may have reduced efficacy the intervention itself
Our Response : We have re-written the limitations, indicating the potential measurement errors, blinding issue, and selection bias.

Reviewer 3 Report

Review on:

 Motivating Diabetic and Hypertensive Patients to Maintain Physical Activity: The Use of Photos and Group Dynamics

In general, the intervention study which is described in this article is a very interesting one. However, its reporting should be more detailed and accurate and its inferences should be more realistic.

Let me start from the conclusion mentioned in the abstract: “Conclusion: This photovoice intervention improved the participants’ physical activity level and physical fitness, particularly in lower limb flexibility and body strength”. This conclusion focusses only on the photovoice, while the intervention included some other components which added to the patients’ knowledge and to their motivation to improve their physical activity. In the discussion, the authors wrote: “This multidisciplinary approach seemed to be a key contributing factor to the success of this photovoice intervention”. It could be that the multidisciplinary approach and not the photovoice by itself contributed to the intervention outcome. However, this speculation is not mentioned. The conclusion, which is mentioned in the abstract might mislead readers who use to read only abstracts (and not the full paper).

Need more accuracy (some examples):

Study design: the authors define their study as “a quasi-experimental study” (row 90) and few rows later they write: “The recruited participants were split into two groups by random number generations: intervention group (IG) and control group (CG) (rows 99-100). They practically describe a controlled clinical trial (not a quasi-experimental). Both randomly selected arms received accelerometers for recording steps.

CBPR: the authors describe: “Community-based participatory method refers to the active engagement of the participants in the photovoice project by using their own life experiences, sharing their views on the selected health topic, propose and develop solutions together with the researchers”. This is only one part of the CBPR principles. Some other parts are described in the discussion instead of mentioned it in the 2.2. Six-Week Intervention. However, in a classical CBPR the participants should take part and have an input to the design of the tools of data collection. In this case, their photos referring to barriers to physical activity could contribute items to the questionnaire about barriers.

More specific methodology: Data analysis section should be much more detailed. The authors used the Generalized Estimating Equations (GEE) models to assess the intervention effect over time on all the outcome variables. In the tables they present the group difference, but it is not mentioned how this difference was calculated.

the use of means for composite variables should be explained: a. using means for description should rely upon a normal distribution of the variables – this is not mentioned.  B. summarizing scales (self-efficacy for physical activity or barriers to exercise) should rely upon their internal consistency (not mentioned).

Personal goals: In the last group meeting (as mentioned in the description of the intervention), participants set up individualized action plans (including goals and timetables) for physical activity in the next four weeks. This does not appear in any other part of the article. It is missing.

By the way, the participants average number of steps taken daily is very high at the beginning (time 0) – this might contribute to the explaining why there are only minor changes after intervention – there is no hint of that in the article.

Author Response

Comments and Suggestions for Authors
R3(Comment 1): In general, the intervention study which is described in this article is a very interesting one. However, its reporting should be more detailed and accurate and its inferences should be more realistic.
Let me start from the conclusion mentioned in the abstract: “Conclusion: This photovoice intervention improved the participants’ physical activity level and physical fitness, particularly in lower limb flexibility and body strength”. This conclusion focusses only on the photovoice, while the intervention included some other components which added to the patients’ knowledge and to their motivation to improve their physical activity. In the discussion, the authors wrote: “This multidisciplinary approach seemed to be a key contributing factor to the success of this photovoice intervention”. It could be that the multidisciplinary approach and not the photovoice by itself contributed to the intervention outcome. However, this speculation is not mentioned. The conclusion, which is mentioned in the abstract might mislead readers who use to read only abstracts (and not the full paper).
Our Response : The abstract was revised to indicate that the multi-component intervention was used: A community-based multi-component intervention (photo-taking, photo-sharing in group discussion, identifying facilitators and barriers to exercise, self-reflection, and action plans) was developed to promote physical activity (PA) among patients with diabetes and hypertension.
Need more accuracy (some examples):
R3(Comment 2): Study design: the authors define their study as “a quasi-experimental study” (row 90) and few rows later they write: “The recruited participants were split into two

groups by random number generations: intervention group (IG) and control group (CG) (rows 99-100). They practically describe a controlled clinical trial (not a quasi-experimental). Both randomly selected arms received accelerometers for recording steps.
Our Response : It is a controlled trial without randomization. We have deleted the word ‘random number generations’ from the sentence.
R3(Comment 3): CBPR: the authors describe: “Community-based participatory method refers to the active engagement of the participants in the photovoice project by using their own life experiences, sharing their views on the selected health topic, propose and develop solutions together with the researchers”. This is only one part of the CBPR principles. Some other parts are described in the discussion instead of mentioned it in the 2.2. Six-Week Intervention. However, in a classical CBPR the participants should take part and have an input to the design of the tools of data collection. In this case, their photos referring to barriers to physical activity could contribute items to the questionnaire about barriers.
Our Response : To avoid misleading, we deleted the description about CBPR.
R3(Comment 4): More specific methodology: Data analysis section should be much more detailed. The authors used the Generalized Estimating Equations (GEE) models to assess the intervention effect over time on all the outcome variables. In the tables they present the group difference, but it is not mentioned how this difference was calculated.
Our Response : The following sentences were added to give more details: (line 240-244) In the GEE models, Time, Group and the interaction term between Time and Group (Time X Group) are independent variables. The coefficient of the interaction term Time X Group estimates the mean difference in the change of the outcome variable over time between the two treatment groups. A significant result in Time X Group indicates a significant differential change in the outcome variable over time between the two groups
R3(Comment 5): The use of means for composite variables should be explained: A. using means for description should rely upon a normal distribution of the variables – this is not mentioned. B. summarizing scales (self-efficacy for physical activity or barriers to exercise) should rely upon their internal consistency (not mentioned).
Our Response : We added the following sentences to explain: Descriptive statistics, including numbers, percentages, means and standard deviation for normally distributed variables, and medians and inter-quartile range (IRQ) for non-normally distributed variables, were performed on demographics, number of steps taken daily and self-efficacy and barriers for doing exercise. Chi-square tests for categorical variables and independent t-test for normally

distributed continuous variables and Mann-Whitney test for non-normally distributed continuous variables to compare the similarity in baseline characteristics between the two groups. Kolomogrov Smirnov Test was used to check the normality assumption of the variables.
We added the Cronbach’s alpha for self-efficacy for exercise, Barriers to Exercise and Chinese Health Literacy Scale (Line 208-209; line 213-214; line 219-220).
R3(Comment 6): Personal goals: In the last group meeting (as mentioned in the description of the intervention), participants set up individualized action plans (including goals and timetables) for physical activity in the next four weeks. This does not appear in any other part of the article. It is missing.
Our Response : We have added the term ‘action plan’ in the Abstract (line 18), Introduction (line 84), Intervention (line 151, 168), Discussion (line 321). Detail discussion about the action plan can be found in line 353-362.
R3(Comment 7): By the way, the participants average number of steps taken daily is very high at the beginning (time 0) – this might contribute to the explaining why there are only minor changes after intervention – there is no hint of that in the article.
Our Response : Agree, the participants’ average number of steps taken daily at baseline is high. We added the following sentence to address this point: (line 332-334) We also noticed that both IG and CG participants’ average number of steps taken per day at baseline was very high, this may partially explained why there was only minor changes after intervention at Week 10.

Round 2

Reviewer 3 Report

As a revised article I’ll focus on the changes made in the article and whether it contribute to understanding the study design and methodology.

1.      The title was changed. It seems that in order to stress the idea of multi-components of the intervention the authors used “The Use of Photos, Group Discussion and Action Planning”, which, in fact, describe the idea of a photovoice. This is only one of the components of the intervention (though it is the main one). Why not use the “photovoice” instead of the long sentence?

2.      In the abstract: The aim of the study, as was revised, support my understanding of the misunderstanding of the idea of a multicomponent intervention. The description of “A

 community-based multi-component intervention (photo-taking, photo-sharing in group discussion, identifying facilitators and barriers to exercise, self-reflection, and action plans)” – all thees components specified in the parenthesis describe the essence of a photovoice, which is only one of the components of the intervention. The intervention included also awareness raising to the contribution of PA to chronic illness management, warm-up stretching exercise, setting individualized PA goals and periodic use of accelerometers (which could serve as a facilitator for PA and not as a measuring tool only). Most of these components are mentioned in rows 82-85. The conclusion, again, specifies the components of the photovoice only. As the intervention included some more activities beyond photovoice only and because the study design unable the isolation of each component, the conclusion should refer to the whole package. However, by giving accelerometers to the control group, at least this component is controlled.

3.      Study design: As an answer to my comments the authors choose to delete the sentence “The recruited participants were split into two groups by random number generations”. This is a very strange answer. If this was the methodology of allocation to IG and CG, it can’t be deleted. If it was not the way of allocation, why was it mentioned in the previous version of the article (what was the real methodology?). In the revised version it was stated that the “Participants were allowed to sign up in the attendance list according to their preference of the start date of the 6-week intervention”. It seems as if the dates were allocated to IG or CG. This should be much clearer.

The rest of my comments were satisfactory answered.

Author Response

Dear Reviewers,

Thank you for your comments and suggestions. We have addressed the comments point-by-point here:

Comment 1

The title was changed. It seems that in order to stress the idea of multi-components of the intervention the authors used “The Use of Photos, Group Discussion and Action Planning”, which, in fact, describe the idea of a photovoice. This is only one of the components of the intervention (though it is the main one). Why not use the “photovoice” instead of the long sentence?

Response:

We have changed the title to ‘Motivating Diabetic and Hypertensive Patients to Engage to Regular Physical Activity: A Multi-component Intervention Derived from the Concept of Photovoice’ to indicate that the photovoice-related activities were the key components of the intervention.

Comment 2

In the abstract: The aim of the study, as was revised, support my understanding of the misunderstanding of the idea of a multicomponent intervention. The description of “A community-based multi-component intervention (photo-taking, photo-sharing in group discussion, identifying facilitators and barriers to exercise, self-reflection, and action plans)” – all thees components specified in the parenthesis describe the essence of a photovoice, which is only one of the components of the intervention. The intervention included also awareness raising to the contribution of PA to chronic illness management, warm-up stretching exercise, setting individualized PA goals and periodic use of accelerometers (which could serve as a facilitator for PA and not as a measuring tool only). Most of these components are mentioned in rows 82-85. The conclusion, again, specifies the components of the photovoice only. As the intervention included some more activities beyond photovoice only and because the study design unable the isolation of each component, the conclusion should refer to the whole package. However, by giving accelerometers to the control group, at least this component is controlled.

Responses:

We agree that the whole intervention is more than photovoice. In the abstract, the aims was revised by changing the contents in the parenthesis. The followings were added:

…(increasing awareness of the importance of physical activity in chronic illness managemet through reading comic books, developing skills to do warm-up stretching exercise, identifying facilitators and barriers to exercise through photo-sharing, supporting self-reflection and development of action plans)…

In the abstract ‘Conclusions’, we have deleted the misleading phase ‘with the use of photos, group discussion and action planing’. The first line of the abstract should be clear enough to indicate the components of the intervention.

Comment 3

Study design: As an answer to my comments the authors choose to delete the sentence “The recruited participants were split into two groups by random number generations”. This is a very strange answer. If this was the methodology of allocation to IG and CG, it can’t be deleted. If it was not the way of allocation, why was it mentioned in the previous version of the article (what was the real methodology?). In the revised version it was stated that the “Participants were allowed to sign up in the attendance list according to their preference of the start date of the 6-week intervention”. It seems as if the dates were allocated to IG or CG. This should be much clearer.

Responses:

Randomization by using random number generations was applied in the first half of recruitment. However, we noted that many participants were hesitated to join the study if the start date of the intervention crashed with their planned events. Therefore we gave up the planned randomization method (using random number generations) but allowed the participants to choose the start date of the intervention. The chosen ‘start date’ determined the allocation to IG and CG in the second half of recruitment.

We do not want to confuse the readers, therefore we determine to remove the original sentence ‘the recruited participants were split into two groups by random number generations’, but only report how we used ‘start date’ to allocate the participants.

In this revised version, we state this clear by adding the following sentences:

In the first few batches of recruitment, we split the recruited participants into two groups by random number generations. However, we gradually found that many participants were hesitated to join the study if the start date of the intervention crashed with their planned events. We therefore allowed participants to sign up in the attendance list…

Grateful if you could consider this revised version.
